# Efficient Path Planing for Articulated Vehicles in Cluttered Environments

**DOI:** 10.3390/s20236821

**Published:** 2020-11-29

**Authors:** Ricardo Samaniego, Rodrigo Rodríguez, Fernando Vázquez, Joaquín López

**Affiliations:** 1Imatia Innovation S.L., Galileo Galilei 64, 15008 A Coruña, Spain; rodrigo.rodriguez@imatia.com; 2Department of Systems Engineering and Automation, School of Industrial Engineering, University of Vigo, 36310 Vigo, Spain; fvazquez@uvigo.es (F.V.); joaquin@uvigo.es (J.L.)

**Keywords:** autonomous robot navigation, mobile robot control, motion planning, articulated vehicles

## Abstract

Motion planning and control for articulated logistic vehicles such as tugger trains is a challenging problem in service robotics. The case of tugger trains presents particular difficulties due to the kinematic complexity of these multiarticulated vehicles. Sampling-based motion planners offer a motion planning solution that can take into account the kinematics and dynamics of the vehicle. However, their planning times scale poorly for high dimensional systems, such as these articulated vehicles moving in a big map. To improve the efficiency of the sampling-based motion planners, some approaches combine these methods with discrete search techniques. The goal is to direct the sampling phase with heuristics provided by a faster, precociously ran, discrete search planner. However, sometimes these heuristics can mislead the search towards unfeasible solutions, because the discrete search planners do not take into account the kinematic and dynamic restrictions of the vehicle. In this paper we present a solution adapted for articulated logistic vehicles that uses a kinodynamic discrete planning to bias the sampling-based algorithm. The whole system has been applied in two different towing tractors (a tricycle and a quadricycle) with two different trailers (simple trailer and synchronized shaft trailer).

## 1. Introduction

In recent years, automation of logistic vehicles has been an important pole of attraction, due to the reduction of costs and operating times that they bring to the industry [1,2,3]. These vehicles carry raw materials and finished goods in productive centers such as factories, workshops or warehouses, providing a flexible material flow.

There are multiple types of vehicles that have been automated to a greater or lesser extent: forklift trucks, reach truck lifts, low lift pallet trucks, stacker trucks, order pickers, tugger trains (also known as “logistic trains”) and others. Among these vehicles, tugger trains (Figure 1) present particular difficulties, due to the kinematic complexity of these multiarticulated vehicles. Moreover, different trailer types present distinct kinematics and follow different trajectories when towed by a tractor vehicle. These systems require the application of specific techniques, both for planning and navigation.

Tugger trains can be used to transport stock material to working locations, because they can supply different stations with different materials efficiently. Figure 2 shows the sketch of a possible scenario in a factory with different stations (production cells). In most of these industrial scenarios, they should drive only in certain areas of a factory site that are marked clearly for the interacting workers.

The first approach to the automation of these vehicles was made by means of simple and insufficiently flexible techniques, such as wire guidance. As computers and sensors evolve, new localization, mapping and navigation techniques are being applied [4], as well as several robotic algorithms [5,6] that provide greater autonomy to these vehicles. These new techniques allow the emergence of more flexible and intelligent vehicles, even adapting routes dynamically to the production and traffic conditions in factories and warehouses [7], thus simultaneously optimizing the routes of several vehicles at once [8]. At the same time, initial systems based on external sensors are evolving for simpler applications, producing low-cost solutions [9].

Several path planning and motion planning approaches have already been applied as part of the navigation system for different AGVs in industrial environments. While the path planning process constructs a path from a starting point to an end point, the motion planning process also includes the set of actions required to follow the path. The sampling-based methods are among the most popular motion planning methods. Within these methods, probabilistic Roadmaps (PRMs) [10] and Rapidly-exploring Random Trees (RRTs) [11] are shown to be quite effective in high-dimensional configuration spaces. In planners developed for different applications, variants of these sampling-based algorithms have been combined in a hierarchical approach using discrete search techniques to bias the search and make it more efficient [12,13]. The main difference with the approach presented in this paper is that we use a kinodynamic planning in the discrete search stage, taking advantage of the specific heuristics and vehicle models for this application. For example, in Figure 2 most of these hierarchical approaches will guide the motion planner towards a path to follow the route 1, because the discrete search techniques that bias the sampling-based planner does not take into account the kinematic and dynamic restrictions of the vehicle. However, this trajectory may be unfeasible for the tugger train, because it cannot execute the left ninety-degrees turn on the narrow corridor of point C. The vehicles can turn from the surrounding area into the corridor (point B) because they have a wider turning area but once in a corridor they do not have enough room to make a ninety degree turn from one corridor to another.

In this paper, we present the application of different technologies for the automation of a logistic tugger train, shown in Figure 1. The techniques hereby presented allow the simultaneous utilization of several trailer elements of different types. When using a single trailer, with the proposed method, it is even possible to perform reverse maneuvers for precise placement in the desired position.

The main contribution of this paper is a novel motion planning technique, especially designed for articulated vehicles, that uses a lattice-based kinodynamic discrete planner to bias a second stage sampling-based algorithm. The contributions include:An efficient lattice-based kinodynamic discrete planning to bias the sampling based algorithm.A sampling based algorithm, based on RRT*, biased by the previous kinodynamic discrete planning.The model, implementation and study on two different kinds of common tugger trains (tricycles and quadricycles) that can tow two kinds of trailers (simple trailers and synchronized-shafts trailers).

The rest of the paper is organized as follows. The first section introduces the work related to this research. The second section describes the towing tractors and trailers including their kinodynamical models used in the planner. The motion planner method is presented in Section 3. The results obtained are summarized in Section 4, together with a study comparing the planner with similar approaches. Finally, Section 5 concludes the paper.

## 2. Related Work

Autonomous path planning and navigation of nonholonomic vehicles has been an active field of research in recent decades [14]. Robotic developments are increasingly moving from small, circular research robots to vehicles that are large and fast, such as cars or trucks. When it comes to motion planning, small circular robots have the advantage that they can stop almost instantly and can turn in place if they are equipped with differential drives, making the path planning problem for them very close to that of a holonomic vehicle. Large vehicles pose many more challenges for path planning algorithms due to their dynamic and spatial characteristics, including a limited turning radius, as well as the fact that they tend to be slender or elongated, rather than circular or almost circular. In addition, besides the rigid vehicles, articulated vehicles such as tugger trains are widely used in logistics, as they provide a very efficient way to supply and dispose different materials to different locations.

Early approaches focused on local planning, where only short term paths are calculated to navigate towards a given goal while avoiding obstacles. Those obstacles can be previously known or detected in real time [15,16]. This local approach is prone to get stuck on convex obstacles, such as cul-de-sacs or other forms of local minima, and it is also incapable of reasoning complex maneuvers for navigating in complicated environments such as sharp turns on narrow corridors or U-turns. Much of the recent work is focused on global path planning taking the robot from the current position to a given goal. The path needs to be compatible with the restrictions posed by the vehicle and the environment [17].

For some navigation systems, the global path is a geometric path. In this case, the path is the reference for a local reactive navigation method that deals with the problem of avoiding unexpected static and dynamic obstacles in a partially known environment. There is an extensive literature on local navigation approaches. Some of them (CVM [16], BCM [18] and dynamic window [19]) formulate the local obstacle avoidance as a constrained optimization problem in the velocity space of the robot. Other well-known local reactive methods are the Artificial Potential Fields [20] and the Vector Field Histogram [21]. However, even if they add mechanisms to avoid getting stuck on convex obstacles, they cannot guarantee to obtain the optimal path.

Other planners compute the global path through the concatenation of minor discrete actions taken from a precalculated control set, forming a state lattice [22]. These actions connect different states of a discretized space, allowing the path planning problem to be formulated as a graph search. This global approach is very challenging in terms of memory consumption and processing time, especially when high resolutions are needed.

The kinodynamic planning algorithms include kinematic and dynamic constraints imposed on the vehicle. For example, in [23] a kinodynamic motion planning with state lattices has been used for the case of a holonomic Automated Guided Vehicles (AGVs) for logistic applications. A Lattice-Based Approach has also been applied in [24] to Multirobot Motion Planning for Nonholonomic Vehicles for industrial intralogistic tasks. The work presented in [25] analyzes the reverse motion of a tractor-trailer with one on-axle hitched semitrailer and develops a full autonomous driving system that enables reverse parking.

Another kinodynamic planning techniques use dynamic programming to search for a solution on a grid of the state space. For example, in [26] the lower level of a two-level algorithm uses the physical model of the robot to find a feasible trajectory. Some trajectory optimization techniques can also include kinodynamic restrictions even though the problem size becomes too big for long time horizons [27].

Some hierarchical solutions separate the problem into a high-level, discrete, geometric planning level and a low-level, continuous, trajectory planning level. For example, in [28] they solve a special shortest path problem on a graph at a higher level of planning, and then use a lower level planner to determine the costs of the paths in that graph.

The sampled-based motion planners are among the more extended planning techniques for robots with complex nonholonomic constraints. For example, the Rapidly exploring Random Tree (RRT) [11] and its optimal variant RRT* [29] expand trees in the state space toward newly sampled states under the application of nonholonomic constraints. However, their planning times scale poorly for high dimensional systems, such as multiarticulated robots.

Some variants of these sampling-based algorithms have been combined or biased using discrete search techniques to guide the search and make it more efficient [12,13,30,31,32]. In [32] the discrete search technique is a graph search on a regular grid. The weight of each grid edge quantifies the roughness of the terrain. The Informed Subdivision Tree (IST) described in [31] integrates the heuristic information in an adaptive subdivision tree. An any-angle search is used in [12] to bias and improve the performance of the sampling-based RRT. In [13] the workspace is decomposed into regions that form a graph that encodes the adjacency of the regions. The graph is searched to obtain sequences of regions that can be explored with sampling-based methods. A similar technique of region decomposition is presented by the same authors in [30].

The approach proposed here also uses a discrete search technique to bias the sampling-based search (Figure 3). The main difference is that for our case (articulated vehicle) the discrete search is a lattice-based kinodynamic planning, to make sure that the robot is going to be able to follow the planned path. None of the previous work uses kinodynamic planning in the discrete search technique. Therefore, they can end up with only unfeasible trajectories in the random search area. They only include the kinodynamic restrictions for the sampling-based algorithms.

## 3. Overview

### 3.1. Problem Formulation

A path can be defined as a set of consecutive states $x_{i}\in X_{free}$ in the free configuration space ($X_{free}=X\setminus X_{obs}$) where the first state is the initial state ($x_{I}$) and the last state is the goal state ($x_{G}$). A motion plan also includes the controls ($u_{i}$) that move the robot from one state to another $\left(x_{i},u_{i}\right)|x_{i}\in X_{free},u_{i}\in U$. The motion planning problem can be seen as the Optimization Control Problem (OCP) that finds the optimal plan according to some cost function *J*. If we define $j(\left(x_{i},u_{i}\right))$ as the cost of taking action $u_{i}$ when the robot is in state $x_{i}$, the problem can be formulated as:(1)$$\min_{u_{i}}J=\sum_{i=0}^{n}j\left(x_{i},u_{i}\right)$$

The minimization is subject to $x_{0}=x_{I}$, $x_{n}=x_{G}$, $x_{i}\in X_{free}$ and u(t)∈U. Considering a continuous configuration space and control space as in [33], a trajectory is a time-parameterized path $\left(x\left(t\right),u\left(t\right)\right)|x\left(t\right)\in X_{free},u\left(t\right)\in U,t\in \left[t_{I},t_{G}\right]$. For this case, the OCP problem can be formulated as:(2)$$\min_{u\left(.\right),t_{G}}J=\int_{t_{I}}^{t_{G}}L\left(t,x\left(t\right),u\left(t\right)\right)\;dt$$

The minimization is subject to $x\left(t_{I}\right)=x_{I}$, $x\left(t_{G}\right)=x_{G}$, $x\left(t\right)\in X_{free}$ and u(t)∈U. Where the Lagrange term L defines the performance measure to be minimized. The evolution of the state x(t) according to the action carried out u(t) is defined by the model of the vehicle:(3)$$\dot{x}\left(t\right)=f\left(x\left(t\right),u\left(t\right)\right)$$

Figure 3 describes the diagram of the algorithm proposed here named LA*-RRT*. The main two steps are the discrete and sampling based searches. In the discrete search technique, the model of the vehicle is needed to define the actions in the lattice. Likewise, for the sampling based search, the model is used to define the actions to move from one previous state to a recently sampled state. The rest of this section is devoted to modeling the vehicles for motion planning. First, we describe the vehicles and then we will define the kinematic model.

### 3.2. Vehicle Description

The work presented in this paper was developed as part of a logistic automation project. The main objective of this project is the development of autonomous robot technologies for scalable and flexible logistic applications. These technologies were tested in two modified Still CX-T towing tractors (Figure 1). In both cases, the CAN bus of the original vehicles was modified in order to send orders to the actuators (acceleration motors and direction).

Two different positioning technologies were used for this work:A scan matching based method. It uses a laser scanner and a SLAM-based previously obtained map [34].A triangulation method based on artificial landmarks. The landmarks are reflectors that are detected by a laser scanner [35].

Both vehicles were equipped with the following sensors:Sick S300 laser scanner in the lower front part, installed inside a custom metallic hood, acting as a bumper (Figure 1). This scanner is used for obstacle detection during local navigation, as well as for localization and global map creation using the SLAM technique.Sick NAV350 laser scanner (Figure 1b), used for localization when the triangulation with optical reflectors method is used.Wheel odometry sensors (optical encoders) in both vehicles, the drive wheel is the same as the steering one.Optical lateral collision sensors, installed at both sides of the vehicles, used as light barriers.

Both towing tractors are also equipped with a rear coupling, connected to an internal hydraulic pump. This actuator is used for lifting and lowering the pallets loaded in all the towed trailers at the stop stations (Figure 4b).

The main difference between both vehicles is in the kinematics, as they are two different versions of the same model. So, the first towing tractor is a quadricycle in which the front left wheel is the drive and steering one (Figure 1a); the second one is a tricycle, with a single frontal wheel acting both as steering and drive (Figure 1b). In the quadricycle vehicle, the front right wheel is a swivel caster. In both vehicles, the two rear wheels are rigid casters.

Two different trailers where used. One of them is used to transport big material gaskets, usually up to 500 kg. This trailer has two shafts, front and rear, synchronized by two metal rings and a diagonal rod. This rod connects both shafts by using a joint on each end (Figure 4a). The second trailer is used for palletized loads, being each pallet mounted on a single trailer. It has a single shaft with two fixed-direction, independent wheels (Figure 4b). Both trailers are coupled to the towing tractor by a bar. In the first trailer type, this bar is fixed to the front steering shaft. In the second one, it is fixed to the trailer frame.

### 3.3. Vehicle Kinematics

Both towing trucks and trailers were modeled. These models were used for the global planner.

In the tugger trains used for this paper, each element, tractor or trailer, may present different dimensions.

#### 3.3.1. Simple Trailer

The dynamic equations of movement for a multiarticulated vehicle with multiple simple trailers are the following:(4)$${\dot{x}}_{k}=\;v_{k}*\cos\left(\theta_{k}\right)\;$$(5)$${\dot{y}}_{k}=\;v_{k}*\sin\left(\theta_{k}\right)\;$$(6)$${\dot{\theta }}_{0}=\;\frac{v_{w}*\sin(\beta_{0})\;}{L_{0}}$$(7)$${\dot{\theta }}_{1}=\frac{v_{0}*\;\sin(\beta_{1})\;}{L_{1}*\cos\left(\alpha_{0}\right)}$$(8)$${\dot{\theta }}_{2}=\frac{v_{1}*\;\sin(\beta_{2})\;}{L_{2}*\cos\left(\alpha_{1}\right)}$$
⋯
(9)$${\dot{\theta }}_{k}=\frac{v_{k-1}*\;\sin(\beta_{k})\;}{L_{k}*\cos\left(\alpha_{k-1}\right)}$$(10)$$\dot{v_{w}}=a$$(11)$$\dot{\varphi }=\omega$$(12)$$\left\{\begin{array}{c} v_{0}=v_{w}*\left(\cos\beta_{0}\;+W_{0}*\frac{\sin\beta_{0}\;}{2L_{0}}\right)\\ v_{k}=v_{k-1}*\frac{c\mathrm{os}\beta_{k}\;}{\cos\alpha_{k-1}\;},\;\;\;0<k\; \end{array}\right.$$(13)$$\left\{\begin{array}{c} \beta_{0}=\varphi \\ \beta_{k}=\theta_{k-1}-\theta_{k}-\alpha_{k-1},0<\;k \end{array}\right.$$(14)$$\left\{\begin{array}{c} \alpha_{0}=\mathrm{atan}\;\left(\frac{{BJ}_{0}}{\frac{L_{0}}{\tan\left(\beta_{0}\right)\;}+\frac{W_{0}}{2}}\right)\;\\ \alpha_{k}=\mathrm{atan}\left(\frac{d_{k}\tan\left(\beta_{k}\right)\;}{L_{k}}\right)\;0<\;k\; \end{array}\right.$$

In these equations:$x_{k}$, $y_{k}$, $\theta_{k}$: Coordinates of the element *k* of the vehicle where the element 0 is the towing tractor. These coordinates are always referred to the central point of the rear shaft of the corresponding element.$v_{k}$: Forward speed of the element *k* of the vehicle.$v_{w}$: Forward speed of the tractor vehicle’s drive wheel.$\beta_{0}$: Angle between the rear shaft of the towing vehicle and the imaginary line that goes from the drive wheel to the instant center of rotation (noted in the drawings of Figure 5 as IRC)$\beta_{k}$, with i>0: Orientation of the towing bar of the element *k*.$L_{k}$: Length of the element *k* of the vehicle. For the tractor elements ($L_{0}$, Figure 5), we use the wheelbase of the vehicle (distance between shafts). For single-shafted trailers, we use the distance between the rear shaft and the joint with the preceding element ($L_{1}$, Figure 5).$\alpha_{k}$: Angle between the rear shaft of the element *k* and the imaginary line that goes from its joint with the next trailer to the instant center of rotation.*a*: Acceleration of the tractor vehicle’s drive wheel.φ: Angle of the steering wheel of the towing tractor.$W_{0}$: Track of the towing vehicle (distance between wheels). This value is 0 for tricycle trailers.$BJ_{0}$: Back-joint distance. Distance from the center of the rear shaft of the towing tractor to the joint with the first trailer.

#### 3.3.2. Synchronized-Shafts Trailer

The dynamic equations of movement used for a multiarticulated vehicle with multiple synchronized-shafts trailers are the following:(15)$${\dot{x}}_{k}=\;v_{k}*\cos\left(\theta_{k}+\gamma_{kB}+\pi /2\right)\;$$(16)$${\dot{y}}_{k}=\;v_{k}*\sin\left(\theta_{k}+\gamma_{kB}+\pi /2\right)\;$$(17)$${\dot{\theta }}_{0}=\;\frac{v_{w}*\sin(\beta_{0})\;}{L_{0}}$$(18)$${\dot{\gamma }}_{1F}=\frac{v_{0}*\;\sin(\beta_{1})\;}{{FJ}_{1}*\cos\left(\alpha_{0}\right)}-{\dot{\theta }}_{1}$$(19)$${\dot{\theta }}_{1}=\frac{v_{1}}{L_{1}}\left(\frac{\sin\gamma_{1B}\;}{\tan\gamma_{1F}\;}-\cos\gamma_{1B}\;\right)$$(20)$${\dot{\gamma }}_{2F}=-\frac{v_{1}\sin\left(\gamma_{1B}\right)\sin\left(\beta_{2}\right)\;}{{FJ}_{2}\cos\left(\alpha_{1}\right)}-{\dot{\theta }}_{2}$$(21)$${\dot{\theta }}_{2}=\frac{v_{2}}{L_{2}}\left(\frac{\sin\gamma_{2B}\;}{\tan\gamma_{2F}\;}-\cos\gamma_{2B}\;\right)$$
⋯
(22)$${\dot{\gamma }}_{kF}=-\frac{v_{k-1}\sin\left(\gamma_{k-1B}\right)\sin\left(\beta_{k}\right)\;}{{FJ}_{k}\cos\left(\alpha_{k-1}\right)}-{\dot{\theta }}_{k}$$(23)$${\dot{\theta }}_{k}=\frac{v_{k}}{L_{k}}\left(\frac{\sin\gamma_{kB}\;}{\tan\gamma_{kF}\;}-\cos\gamma_{kB}\;\right)$$(24)$$\dot{v}=a$$(25)$$\dot{\varphi }=\omega$$(26)$$\left\{\begin{array}{c} v_{0}=v_{w}\left(\cos\beta_{0}\;+W_{0}*\frac{\sin\beta_{0}\;}{2L_{0}}\right)\\ v_{k}=v_{k-1}\frac{\cos\beta_{k}\;*sin\gamma_{Bk-1}*sin\gamma_{Fk}}{\cos\alpha_{k-1}\;*sin\gamma_{Bk}},\;\;\;0<k\; \end{array}\right.$$(27)$$\left\{\begin{array}{c} \beta_{0}=\varphi \\ \beta_{i}=\theta_{i-1}-\theta_{iS}-\alpha_{i-1},0<\;i<k\; \end{array}\right.$$(28)$$\left\{\begin{array}{c} \alpha_{0}=\mathrm{atan}\left(\frac{{BJ}_{0}}{\frac{L_{0}}{\tan\left(\beta_{0}\right)\;}+\frac{W_{0}}{2}}\right)\;\\ \alpha_{i}=\mathrm{atan}\left(\frac{L_{i}-BJ_{i}\;\left(\frac{\tan(\gamma_{Bi})}{\tan(\gamma_{Fi})}-1\right)}{L_{i}\tan\left(\gamma_{Bi}\right)}\right)\;0<\;i<k\; \end{array}\right.$$

In these equations:$x_{k}$, $y_{k}$, $\theta_{k}$: Coordinates of the element *k* of the vehicle. Element 0 is the towing tractor. These coordinates are always referred to the central point of the rear shaft of the corresponding element.$v_{k}$: Forward speed of the element *k* of the vehicle.$\gamma_{kF}$, $\gamma_{kB}$: Angle of the front and rear (back) shaft of the trailer *k*.$v_{w}$: Forward speed of the tractor vehicle’s drive wheel.$\beta_{0}$: Angle between the rear shaft of the towing vehicle and the imaginary line that goes from the drive wheel to the instant center of rotation (noted in the drawings as IRC)$\beta_{k}$, with k>0: Orientation of the towing bar of the element *k*.$L_{k}$: Length of the considered element. For the tractor elements ($L_{0}$, Figure 5), we use the wheelbase of the vehicle (distance between shafts). For synchronized-shaft trailers, we use the distance between the front shaft and the joint with the next element ($L_{1}$, Figure 6).$FJ_{k}$: In the synchronized-shafts trailer, this is the length of the bar connecting with the element *k*, which goes from the center of the front shaft to the joint with the preceding element.$\alpha_{k}$: Angle between the rear shaft of the element *k* and the imaginary line that goes from its joint with the next trailer to the instant center of rotation.*a*: Acceleration of the tractor vehicle’s drive wheel.φ: Angle of the steering wheel of the towing tractor.$W_{0}$: Track of the towing vehicle (distance between wheels). This value is 0 for tricycle trailers.$BJ_{0}$: Back-joint distance. Distance from the center of the rear shaft of the towing tractor to the joint with the first trailer.$\theta_{kS}$: Angle of the front rod of trailer *k*, which links it to the rear vehicle k−1.

As observed in Figure 6, the instant value of $\gamma_{kB}$ is only dependent of $\gamma_{kF}$. The previous equations show that the calculation complexity for $\gamma_{kB}$ is very high, requiring many computational resources to be solved in real-time. For this reason, an empiric approach was taken, linearizing the values of $\gamma_{kB}$ in the working range of the shafts, that goes between $60^{\circ }$ and $120^{\circ }$. Figure 7 shows that the working range is within the area that can be linearized (55–$140^{\circ }$).

## 4. Motion Planning

The solution proposed here is a hierarchical technique that combines a kinodynamic discrete planning (lattice-based A*) with a continuous sampling technique (RRT*). The kinodynamic planner obtains a first plan that defines the area to focus the growth of the RRT* trees. In other words, it is a two phase planner; in the first phase, a kinodynamic discrete planner finds a path that is used to define the area where to apply, in the second phase, a continuous sampling technique. The kinodynamic discrete planning is carried out in a reduced dimension space (just for the towing tractor). Therefore, the time to obtain the first path is negligible compared to the total planning time. Then, we apply a mask to the map around this first path, so that in the second planning, sampling-based, only the states within this mask can be sampled. If the heuristic applied to the search-based algorithm is admissible, the optimal path will be included within the mask.

The aforementioned mask will be obtained by moving a kernel through all the points that form the discrete path. This technique is based on two assumptions:The trailers of an articulated vehicle follow a similar route as the tractor unit that tows them. However, each element of the vehicle slightly “shortens” the path followed by the preceding element, as shown in Figure 8. Therefore, the greater the number of trailers, the greater the deviation from the route followed by the last towed element with respect to that followed by the tractor unit. The difference between the routes traced by contiguous elements will depend on the dimensions of these elements and the specific mechanics of the trailers. In Section 3.3, two different types of trailer were analyzed concluding that they present different deviations with respect to the trajectory of the element that precedes them.For most of the logistic applications, the vehicles should be able to navigate through the working space to reach all the pick-up and drop-off positions. This means that all corridors and intersections through which the towing vehicle can move without trailers are also practicable for the same head with trailers, although surely following a different route, “opening” the path in the curves in order to save space for the trailers. This assumption is true for virtually all industrial environments in which a tugger train is used.

The smaller the kernel used to create the mask around the first phase path, the smaller will be the sampling space for the second planning phase and the faster the planning. However, an excessively small kernel can prevent one from obtaining a feasible path by reducing too much the vehicle maneuverability. It is therefore necessary to find the right size of the kernel. This can be done empirically, running a battery of simulations for a given articulated vehicle configuration and progressively decreasing the kernel size. It is possible to use a rectangular kernel whose aspect ratio (length/width) is the same as the tractor unit. This approach, however, requires rotating the kernel as it moves along the path. In this document we will apply kernels with a circular shape, for simplicity.

From the trailers described in Section 3.3, we can conclude that using complex trailer geometries (synchronized shafts) allows for a reduction on the size of the kernel. This is due to the fact that simpler geometries tend to have a greater deviation between the path of two consecutive elements of the vehicle (see Section 3.3). In addition, the closer the path returned by the discrete search algorithm (A*) is to the final path, the more it will be possible to reduce the kernel and consequently, reduce the search space. For this reason, it is convenient to carry out a discrete planning with kinodynamic constraints, since it will obtain a path closer to the final one than a simple holonomic planning with two degrees of freedom (x,y). In the present research we use lattice-based planning. The control set used to generate the lattice must be adapted to the specific dynamics of the vehicle.

In the results presented in this paper, the following configuration was used:Search-based planning: a lattice-based ARA* planner with ϵ=1 was used. The planning is carried out in a three dimensional space (x,y,r). The control set used to generate the lattice was adapted to the dynamics of the tractor unit. An admissible heuristic (euclidean distance) was used for this stage.Sample-based planning: it was carried out using a modified version of the RRT* algorithm, including the following features:
-Given the impossibility of analytically solve the vehicle motion equations for the evolution between two given states, the kinodynamic model presented in Section 3.3 is used to generate new states. Thus, the initial state is propagated by sampling random controls of the vehicle, choosing the set of controls that end in the closest position to the proposed state within a distance window.-The state rewiring step is not used. This step consists of evaluating whether any of the neighbors of a new state $q_{new}$ can disconnect from its current parent and connect to $q_{new}$, leading to a shorter or lower cost path. The reason for discarding this step is the impossibility of precisely reaching a given state by sampling random controls from a nearby state. Thus, various approaches to this problem were tested. These approaches were based on searching for states close enough so that they could be considered equivalent. However, the sampling time required to obtain these states is very high. The tests showed that better results are obtained by spending that time sampling new states, instead of trying to rewire the existing ones.-The tree pruning step of RRT* is not used. This step is in charge of discarding parts of the tree taking into account the costs associated with each state. When planning for an articulated vehicle, states that a priory have a higher cost could be those that allow the vehicle to take a curve that is wide enough, so that the trailers do not generate a collision.-The version of RRT* used in this research will always run for a fixed allocated amount of time, even if a solution is found. This way, the path can be progressively improved until the given time is up.

Therefore, the proposed method includes three steps:Obtain an initial heuristic path by using search-based planning (lattice-based A*).Generate a mask around this initial path and apply it to the map.Obtain the final path using a sampling-based planner (RRT*) on the masked map.

### 4.1. Obtaining an Initial Heuristic Path by Using Search-Based Planning

For the search-based planning step, the SBPL library [36] implementation of the ARA* algorithm is used, with a value of ϵ=1. According to the implementation used in this research and for this epsilon value, this is in practice analogous to using A*. The Euclidean distance is used as the heuristic to bias the search.

For the interconnection of the states in the control space, a lattice planner is used. The lattice is formed by combining a set of controls compatible with the dynamics of the vehicle. The implementation of this method follows the description in [22]. This planner generates a feasible path for the tractor unit without trailers. This planning is carried out in a space with reduced resolution, since it is not necessary to obtain a high resolution path but only an estimate of the optimal path to generate the mask around it. The generated control set is shown in Figure 9.

### 4.2. Generating the Mask and Applying It to the Map

Once the list of points that form the optimal path for the tractor unit has been obtained, the mask to be applied on the state space can be generated. This mask is generated by displacing a specific circular kernel along the path. The parameters that influence the size for this kernel are:Tractor size.Number and order of trailers.Size of each trailer.

To obtain the kernel size, an approach following to the next three steps has been implemented:Getting the length of the longest of all the elements of the train and the width of the widest of all the elements of the train.Drawing a rectangle with the two dimensions obtained in the previous step.Drawing the circumference that circumscribes the rectangle of the previous step.

In order to speed up the sampling-based search stage, by reducing the search space even more, the size of this kernel can be shrank. To avoid shrinking it too much for the final vehicle to maneuver between obstacles, an iterative empirical approach can be used. It starts with a circular kernel with the diameter of the aforementioned circumference and then runs a battery of road tests for the train configuration considered. If the results of the paths obtained is not satisfactory, the kernel size is extended and the same battery of tests is executed again. This procedure is repeated until the minimum kernel size for which routes can be safely planned is determined.

### 4.3. Obtaining the Final Path Using a Sampling-Based Planner

The implementation of this step is based on the RRT* functions available on the OMPL library [37]. We named this RRT* version with kinodynamic restrictions as “KinoDynamic RRT* algorithm” or simply KD-RRT*. The control parameters of the tractor unit and a kinodynamic model of each of the elements of the train (head and trailers) is used for the generation and connection of states within the basic operation of RRT*. Since we are dealing with an articulated vehicle, it has been considered that when calculating the path length, only the trailer to be positioned (generally the last one) is taken into account.

The changes proposed for this algorithm are mainly included in two steps of its execution: sampling and connection. At the end of each connection stage, the new sampled node is added to the tree and the distance between this node and the goal is obtained. If this new node is close enough to the objective, it is considered that the goal has been reached. However, even after a first path is found, the execution will continue until the allocated time is up. This allows for the improvement of the solution while there is time remaining.

#### 4.3.1. Sampling

The first step in the KD-RRT* algorithm is the random generation of a new (x,y) pair in the subspace of the state space ($x,y,r_{0},r_{1},\ldots ,r_{n}$), where *n* is the number of trailers and x,y are within the mask. We name this state $q_{rand}$ (Figure 10).

If states were simply randomly sampled within all the environment, most of them would fall out of the previously generated mask, wasting a lot of time. To avoid this and to obtain $q_{rand}$ efficiently, a list of the cells of the occupancy map covered by the mask is generated and random integer indices between 1 and $n_{cells}$ are sampled, where $n_{cells}$ is the total number of cells covered in the list. This allows to always obtain a valid $q_{rand}$ state with negligible CPU time consumption.

The next step is to find the state of the pre-existing tree closest to $q_{rand}$ (which we name as $q_{near}$). In addition, a random control is generated starting from $q_{near}$. The final state produced after applying this new control over $q_{near}$, called $q_{new}$ is the candidate state to be added to the tree (Figure 10).

It is worth mentioning that, unlike in the lazy algorithms [38], in KD-RRT* each $q_{new}$ state is validated during the propagation. That is, each time a new state is generated, it is also validated to make sure that it is a feasible state and the vehicle in that state does not collide with an obstacle or produces a self-collision.

To efficiently check the absence of collisions, the possible orientation values are reduced to *p* discrete orientations, and the footprint of each of the elements is precalculated for those *p* values. The value of *p* is defined as a function of the resolution of the occupation map, as defined in [39]. Thus, when checking the collision of a element of the vehicle, the orientation of that element is approximated by the closest value in the set of *p* orientations. Using that orientation, the precalculated footprint is applied. This speeds up the algorithm’s execution, because it is not necessary to calculate the footprint for the current orientation in real time.

The method used to check the absence of self-collisions between the different elements of the articulated vehicle is based on checking that the difference between orientations of two consecutive elements above a known safe value. For the results shown in this paper, a value of 60${}^{\circ }$ was used as this threshold angle.

#### 4.3.2. Connection

The next step is to select a state from the current tree to which we will connect $q_{new}$. Note that, despite $q_{new}$ was generated from $q_{near}$, it is possible that there exists a lower cost path to $q_{new}$ from other states of the tree. To overcome this possibility, the *m* neighbors of $q_{new}$ {$q_{i},\forall i\in (0,m$)} are obtained, and a possible control is calculated from each of them. Due to the use of random controls, it is unlikely that none of them reach $q_{new}$. Instead, each one will end in a different new state named $q_{i\;new}^{\prime }$ (Figure 11). However, since the only purpose of $q_{new}$ is to select the pre-existing node to be expanded, this is acceptable. To validate the $q_{i\;new}^{\prime }$ states we follow the same process used with $q_{new}$ on the sampling stage described above.

The next step is to evaluate which one of the states $q_{i\;new}^{\prime }$ has a lower global cost. The one with less cost is included in the tree, connected to its respective parent. This global cost will be the sum of the node’s parent cost ($q_{i}$) and the expansion cost from the parent to the new one. To obtain this expansion cost, the following expression is used:(29)$$cost\left(q_{i}\to q_{i\;new}^{\prime }\right)=\left(t+k_{1}*\varphi +k_{2}*\left|\theta_{0}-\theta_{N}\right|+k_{3}*\left|\theta_{0}-\theta_{g}\right|\right)*K_{m}$$
where:*t* is the traveling cost between $q_{i}$ and $q_{i\;new}^{\prime }$ (see below).$k_{1}$ is a parameter that penalizes the execution of sharp curves (high values on the orientation of the steering wheel). It is responsible for discouraging unnecessary excessive actions on the steering of the tractor.$k_{2}$ it is a parameter that penalizes the difference in alignment between the tractor element and the last towed element. It is responsible for discouraging actions that involve unnecessarily sharp maneuvers.$k_{3}$ is a parameter that penalizes the difference in orientation between the driving element and the direction towards the goal. It helps to steer the vehicle towards the target, especially in environments with little presence of obstacles.$k_{m}$ (*m* for maneuver) presents a different value for each set control of the lattice. It is usually one for straight forward maneuvers and a higher value for curves, presenting the maximum value for maneuvers performed in reverse. It helps to favor the preference for straight and forward trajectories.$\theta_{g}$ is the orientation needed to direct the tractor element towards the goal from the current position.

A popular choice for the traveling cost is to use the distance between the position variables (x,y) of those states. However, this only makes sense in holonomic vehicles; that are able to perform movements in all directions. For the kind of vehicles considered in this research, curves with dynamic restrictions are commonly used, so it is more accurate to evaluate the exact length of the path that the tow vehicle traces from one state to another. According to the model described before, the kinematic equations of uniformly accelerated motion have been used to calculate the length of each section between two consecutive states.

## 5. Results

To validate the proposed method, a series of tests have been carried out under different conditions. The following planning methods will be compared in the tests included in this section:RRT*. This is a single-phase motion planner that uses RRT*.ARA*-RRT*. This is a two-phase motion planner. In the first phase, an holonomic ARA* finds a geometrically feasible path for a vehicle consisting only of the tractor unit without taking into account its kinodynamic restrictions. That path is used in the second phase to bias the growth of RRT*.LA*-RRT* (Lattice A*-RRT*). This is the two-phase motion planner proposed in this paper. The first phase uses a lattice-based A* to find a first kinodynamic feasible path for the tractor unit. Then, RRT* is used to generate a path for the articulated vehicle in the masked area surrounding the first path.

All the tests were carried out in the same map, with the vehicle starting at the same initial position but varying the goal positions. They were executed in a regular computer, equipped with an Intel(R) Core(TM) i7-8550U CPU processor and 16 Gb of RAM.

In a tugger train, the objective is usually positioning the items transported at a certain point, allowing the tractor unit to maneuver within the available space to achieve this objective. Therefore, all the indicated goal positions correspond to target positions for the center of the rear axis of the last trailer.

The vehicle used for this set of tests is a tugger train formed by a tricycle tractor unit with two simple, single axle, trailers. For the map, an environment formed by corridors and rooms that is suitable for the traditional use of a tugger train has been chosen. Dimensions of the map are 54.1 m length and 43.3 m width, with a cell resolution of 0.1 m. The dimensions of both the tractor and the trailers were 1.6 m long and 0.8 m wide. A maximum linear speed of 5 m per second and a maximum driving wheel angular speed of 45 degrees per second were considered for the vehicle kinematics, with a maximum driving wheel absolute angle of ±90 degrees. For the mask, we used a circular kerner with a diameter of 1.6 m, equal to the tractor length.

The different final positions chosen can be seen in Figure 12, where the vectors indicate the desired final position and orientation for the last trailer of the train. Each test will focus on evaluating different characteristics of the algorithms:The goal in the first test is in a position right after the vehicle leaves the room where it starts. This is the simplest planning problem, in which the algorithm is only required to be able to find a path that allows the vehicle to leave the room in a safe way for the tractor unit and its trailers.For the second test, the goal is located farther from the initial position. In order to reach the goal, the vehicle needs to carry out a 90${}^{\circ }$ turn to the right.For the third test, the goal is located in a room far from the initial one, which also requires a 90${}^{\circ }$ maneuver, performed just through a narrow entrance.In the fourth test, reaching the goal position requires a complex maneuver around a central core to be able to position the trailer in it in the desired direction.

Multiple planning attempts were executed for each method and goal. In all cases, the algorithms were given sixty seconds for planning. For the cases where no path was found after the sixty seconds, a “no solution found” was set for that test. In all cases, once an initial solution was obtained, the algorithms were allowed to spend the rest of the time to refine it and obtain lower-cost solutions. The time elapsed until obtaining an initial solution, the cost of the initial solution and the cost of the final solution were recorded. Table 1 shows the average and standard deviation achieved for the previous values as well as an efficiency indicator that reflects the percentage of times in which the algorithm was able to find a valid path, within the time provided. For the two stage methods (ARA*-RRT* and LA*-RRT*), the time shown in the table includes the total time spent by both stages. In all cases, the execution time for both ARA* and Lattice-A* was almost negligible compared to that used in the RRT* phase (always under half a second).

### 5.1. Goal 1

The paths and RRT* trees obtained by the three algorithms when planning a path from the starting position to the first final position are shown in Figure 13. In order to present the results in more detail, only the portion of the map between the start and goal positions is shown. The path found by ARA*/LA* is shown in magenta, which is used to define the search area (mask) for the RRT* phase. The map outside the mask created around the ARA*/LA* path is shaded in yellow in Figure 13. The states tree sampled by RRT* are sown in blue and the final path for the towing tractor is represented by a green line.

The results on Table 1 show that the RRT* algorithm without any type of mask was not able to find a valid solution in any of the tests carried out (Figure 13a). The high dimension of this problem is the main reason for this result. RRT* spends all the time sampling combinations of movements within the room but is not able to find a combination of actions to safely move the tractor unit and the two trailers through the narrow exit.

Both the ARA*-RRT* (Figure 13b) and the LA*-RRT* methods (Figure 13b) manage to find a solution within the given time in 100% of the tests. Because of the kinodynamic restrictions, the first stage path of LA*-RRT* presents smoother curves that the one obtained by ARA*-RRT*. Still, they obtain final solutions with similar cost and find the first solution in similar times. The time variations between consecutive attempts for the same method were more significant than the differences between both methods, as reflected in the standard deviations indicated in Table 1.

It is important to note that, with an unlimited search time, RRT* is able to find trajectories to leave the initial room and, eventually, reach the goal. Figure 14 shows the actions evaluated by RRT* (Figure 14a) and the proposed method (Figure 14b) when the start and end positions are located in opposite corners of the map. The problem is that RRT* tends to sample all the available space, wasting a lot of computational effort in rooms far from the optimal path, while the proposed method concentrates its efforts around the optimal path, thus allowing one to obtain routes in a shorter period of time.

### 5.2. Goal 2

The results obtained for this second scenario are very similar to those obtained for the first one. Again, RRT* is not able to find a solution to the problem within the given time, because it is not able to find a way out of the initial room. Therefore, RRT* is not included in Figure 15. The differences between ARA*-RRT* (Figure 15a) and LA*-RRT* (Figure 15b) are not relevant. As a matter of fact, the differences between executions of both methods are similar to the differences obtained between successive executions of the same method. This is due to the fact that the execution times of ARA* and LA* are negligible compared to those of RRT* in both methods and also none of the masks obtained present a significant advantage. The only difference found is that the proposed method produces smoother and more natural-looking paths, thanks to the kinodynamic restrictions. Another consequence of this difference is that the path returned by the discrete search algorithm (LA*) is closer to the final path. This allows for a further reduction of the kernel and consequently, a reduction in the search space. Nevertheless, here we have used the same kernel to compare both algorithms.

### 5.3. Goal 3

In this scenario, the vehicle is intentionally forced to go through a narrow entrance in the last room, while making a ninety degree turn. No results for RRT* without a mask are presented, given the impossibility of this algorithm to find a solution that allows the vehicle to leave the initial room.

For this case, it can be concluded from the results in Table 1 that the proposed method (Figure 16b) is able to find a solution within the given time in 100% of the cases. That is because the use of lattices manages to find a first route that deviates previously from the wall, enough to be able to pass through the entrance at the proper angle.

However, since ARA*-RRT* does not take into account the kinodynamic restrictions of the vehicle, it provides a first path and mask close to the wall. Within that mask, only a few trajectories that start on the very left side are feasible for the trailer. Otherwise, the vehicle is not able to make a safe turn without bumping into the right side. Therefore, there are only a few safe trajectories within the mask, which are difficult for RRT* to find. That is why this algorithm only manages to find the solution in 20% of the cases, while in the rest of them the path obtained ends at that entry (Figure 16a).

### 5.4. Goal 4

This scenario forces the vehicle to perform a detour maneuver to face the target position in the appropriate direction. Otherwise, it is impossible for the vehicle’s kinematics to make the ninety-degree turn that would be necessary in the central corridor. Under these conditions, the mask provided by ARA* forces the vehicle to make that impossible turn, so the RRT* stage fails 100% of the time in trying to find a solution. Figure 17a shows that, although the towing vehicle can make a turn, there is no space available within the mask to correctly position the last trailer. However, LA*-RRT*, manages to find a valid solution in all cases, within the given time.

For this scenario, as in the previous one, no tests with RRT* are shown because the method fails to find a way out of the first room.

For the planning tasks, the tests carried out with the real vehicles described in Section 3.2 present the same results as in simulation because planning is carried out with the vehicle models as described above.

## 6. Conclusions

In this paper, a new motion planning algorithm named KD-RRT* has been introduced. This solution is specially adapted for articulated logistic vehicles and uses a kinodynamic discrete planning to bias a sampling-based algorithm. The mask obtained by the discrete planning focuses the growth of the RRT* tree, improving the performance of the sampling-based motion planner. Unlike previous approaches, the use of a kinodynamic discrete planner instead of other nonkinodynamic planners such as A* or Theta*, makes it more suitable for this kind of articulated logistic vehicle.

This new approach has been evaluated in two modified Still CX-T towing tractors (Figure 1) with two different trailers. The first one has two shafts, front and rear, synchronized by two metal rings and a diagonal rod. This rod connects both shafts by using a joint on each end (Figure 4a). The second trailer has a single shaft on the front with two fixed-direction independent wheels (Figure 4b).

The results shown in Section 4 were obtained in a series of tests conducted on a map of 54.1 m width and 43.3 m height, with a cell resolution of 0.1 m. The vehicle configuration used was a tractor with two identical simple-shaft trailers. The dimensions of both the tractor and the trailers were 1.6 m long and 0.8 m wide.

The performance of the KD-RRT* planner was compared with two planners: a pure sampling-based planner (RRT*) and a hierarchical planner ARA*-RRT* that uses a discrete planner (ARA*) to direct the growth of the RRT* tree. This comparison was made using a lattice A* planner for the search based planning step of KD-RRT* and RRT* for the sampling-based (thus, we denoted it also as LA*-RRT*). The results show that the LA*-RRT* approach improves considerably the performance of the RRT* with a great reduction of the planning time. This is expected because there is a great reduction of the search space and samples are generated on the vicinity of a first solution proposed by the discrete planning.

On the other side, the results compared to those produced by ARA*-RRT* show that, for this kind of problem, the algorithm proposed here outperforms ARA*-RRT*. Both hierarchical solutions combine a discrete planner with the same sampling-based planner (RRT*). The discrete planner ARA* produces a plan faster than the kinodynamic discrete planning (LA*). However, sometimes the ARA* misleads the random search towards unfeasible solutions because it does not take into account the kinematic and dynamic restrictions of the vehicle. Since the planning time of these discrete planners is negligible compared to the RRT* stage planning time, the small increase in time spent by the discrete planner is negligible with respect to the benefits obtained.

The analyzed methods were tested in several planning problems within the same map but with goals located in different positions. For the simplest problems where no complex maneuvers were needed and there was plenty of space both hierarchical methods perform well. However, as we increase the complexity of the scenario the ARA*-RRT* method starts failing to find a feasible path because none of them is included in the search area obtained in the first step. For example, in the last scenario (Figure 17a) where LA*-RRT* performs fine, the ARA*-RRT* is unable to find a feasible path in reasonable time (Section 5.4). The reason for this behavior is that when adding the kinodynamic restrictions, there is no feasible path in the space around the path generated by the ARA* algorithm.

## Figures and Tables

**Figure 1 sensors-20-06821-f001:**
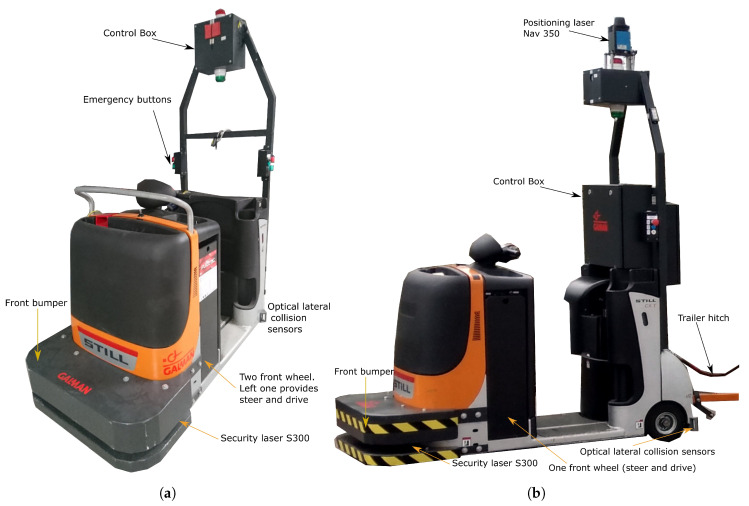
Vehicles used in this research: (**a**) quadricycle tugger; (**b**) tricycle tugger.

**Figure 2 sensors-20-06821-f002:**
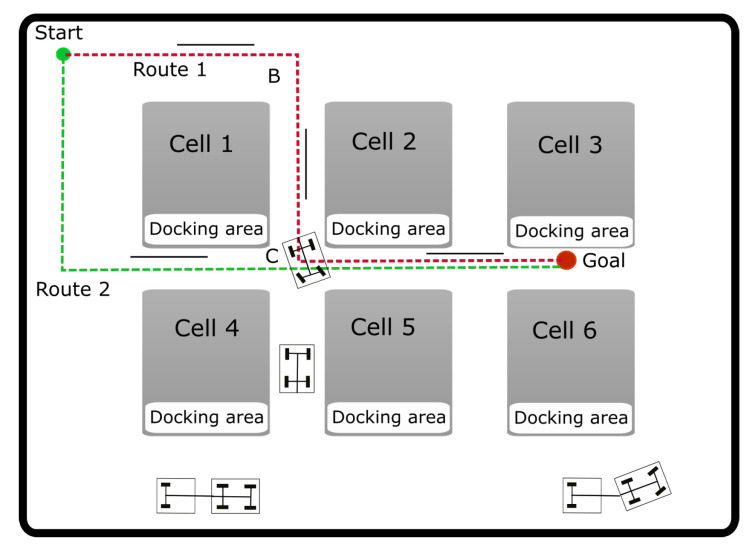
Tugger train working area. There is a dropping/pick-up point in each cell. Tugger trains can turn from the surrounding area into the corridors but once in a corridor they cannot turn ninety degrees into other corridor.

**Figure 3 sensors-20-06821-f003:**
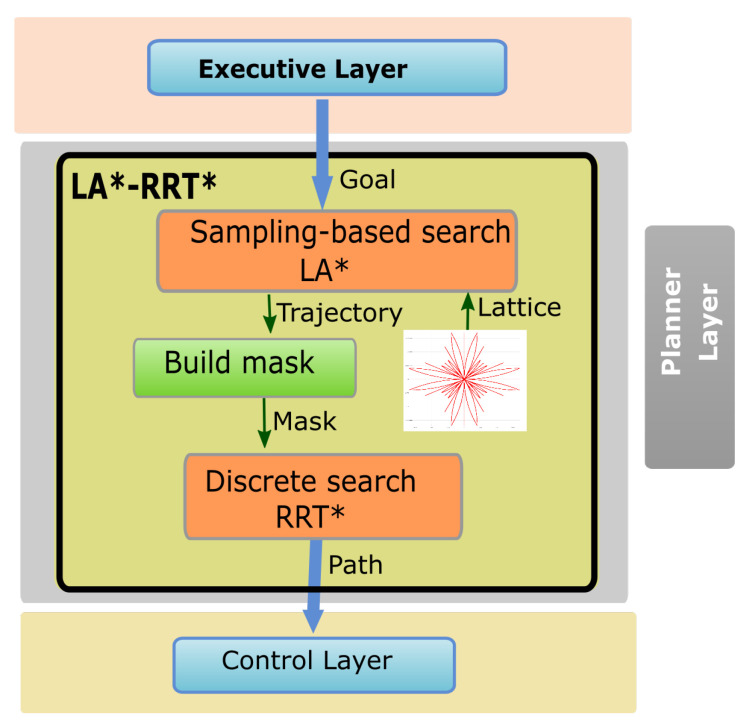
LA*-RRT* block diagram.

**Figure 4 sensors-20-06821-f004:**
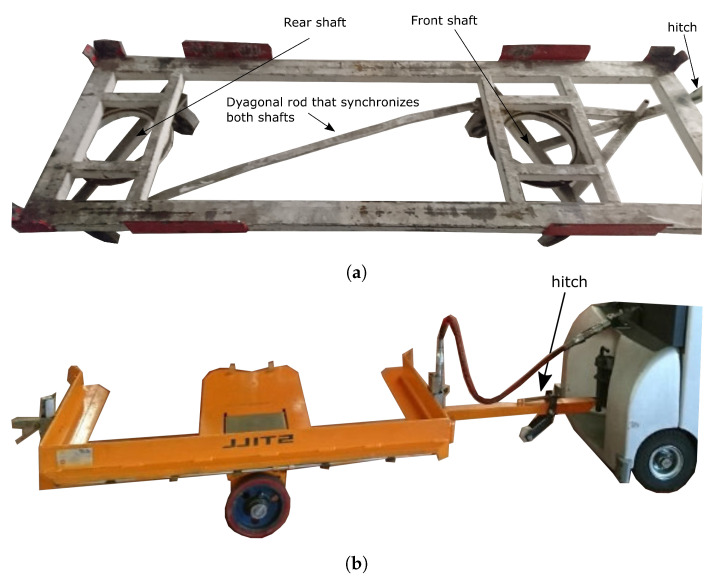
Trailers used in this research: (**a**) synchronized shafts trailer; (**b**) single-shaft pallet trailer.

**Figure 5 sensors-20-06821-f005:**
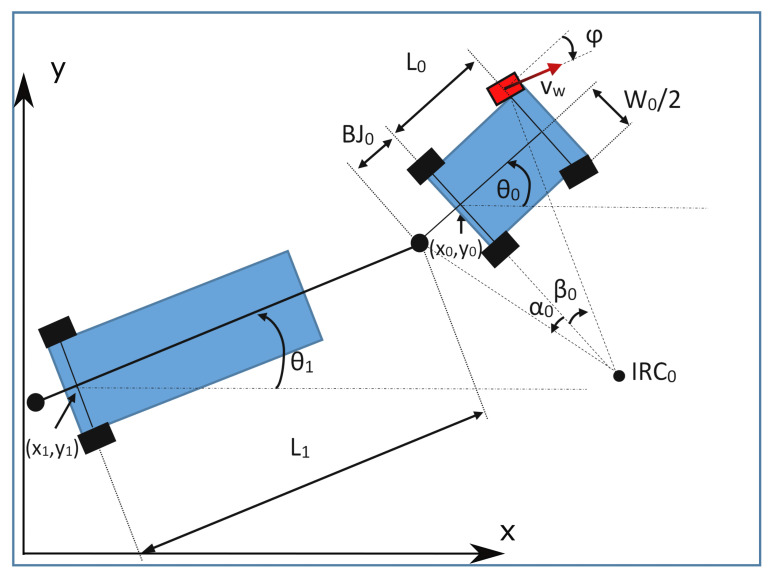
Simple trailer vehicle.

**Figure 6 sensors-20-06821-f006:**
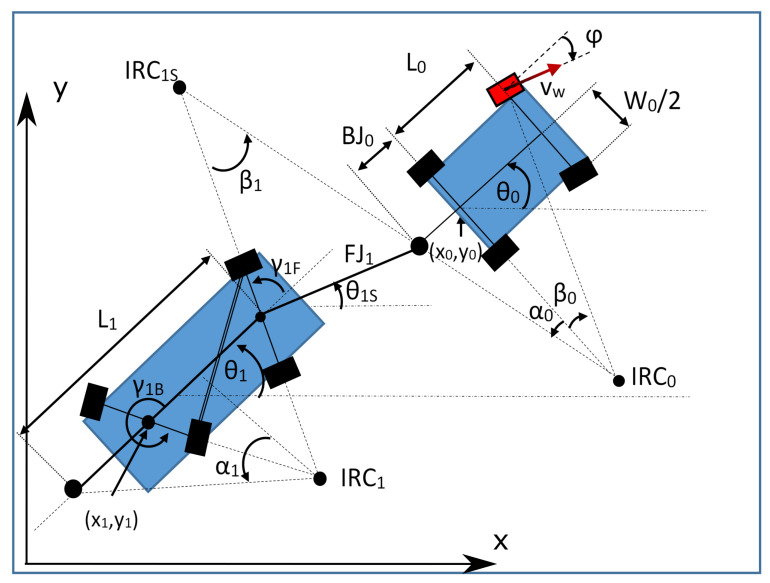
Vehicle with synchronized-shafts trailer.

**Figure 7 sensors-20-06821-f007:**
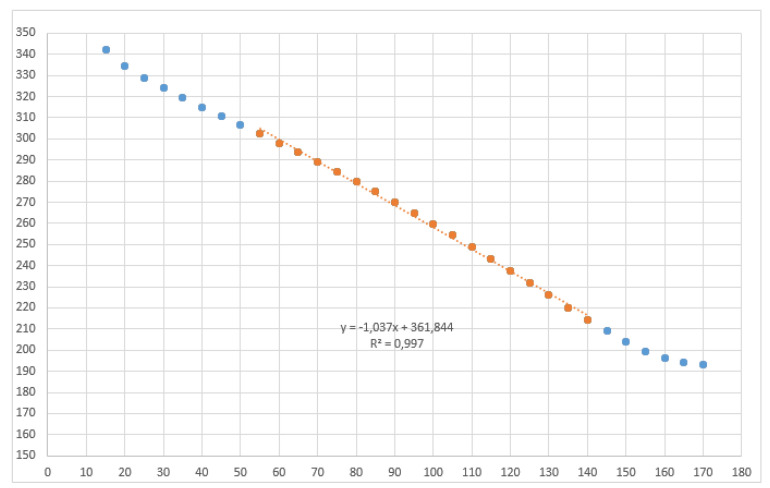
Empirical adjustment of $\varphi_{kB}$ (ordinates) from $\varphi_{kF}$ values (abscissas). Values are shown in degrees.

**Figure 8 sensors-20-06821-f008:**
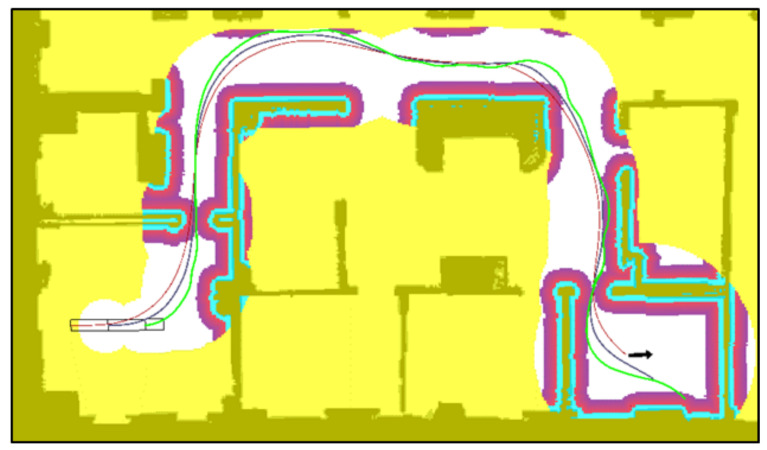
Trajectories followed by the tractor (green) and each of the two trailers (blue and red).

**Figure 9 sensors-20-06821-f009:**
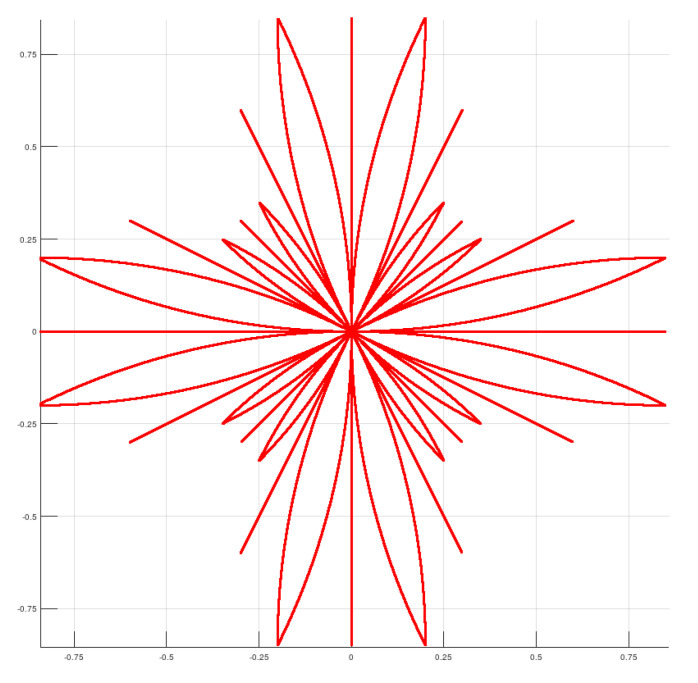
Control set used for the search-based planner. Axis values are shown in meters.

**Figure 10 sensors-20-06821-f010:**
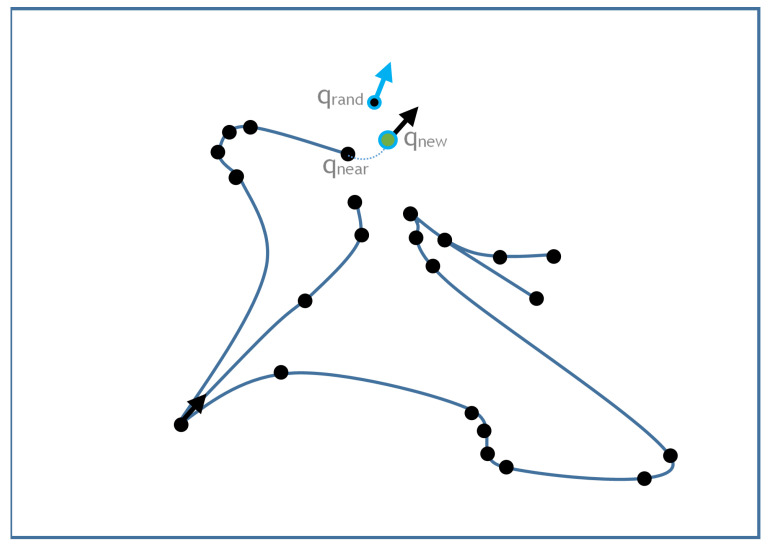
Sampling step.

**Figure 11 sensors-20-06821-f011:**
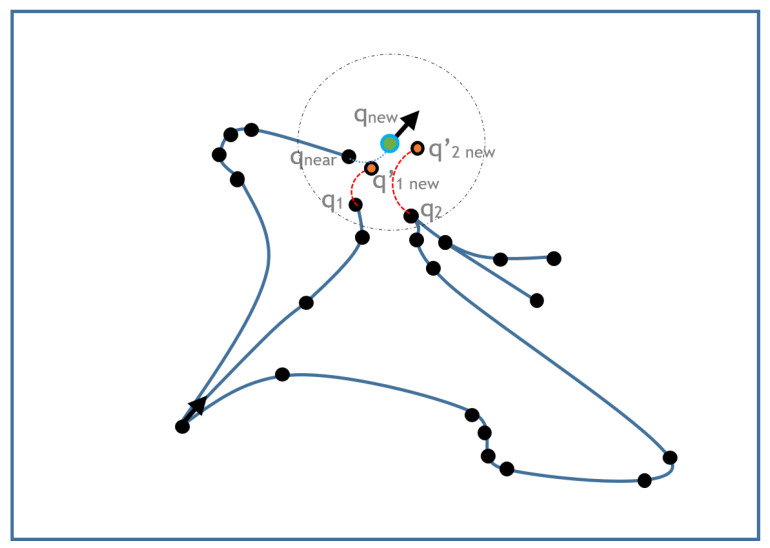
Connection step.

**Figure 12 sensors-20-06821-f012:**
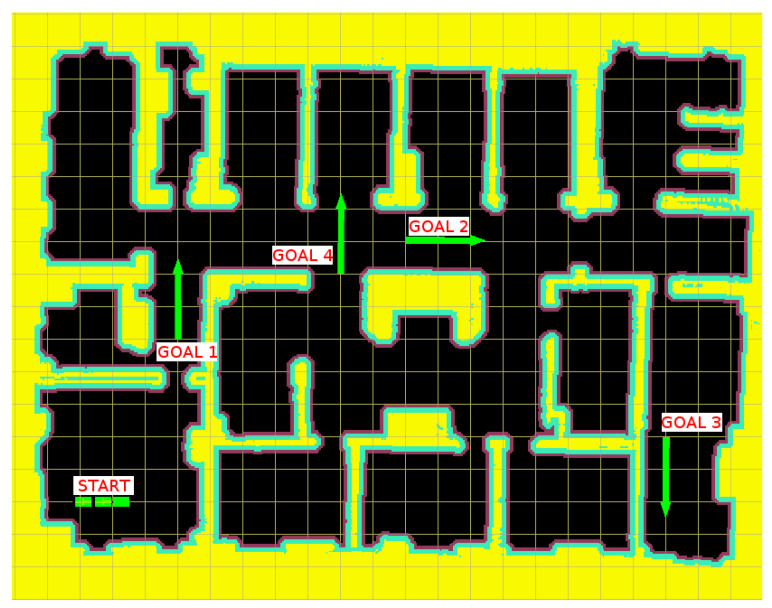
Goals selected for each set of tests.

**Figure 13 sensors-20-06821-f013:**
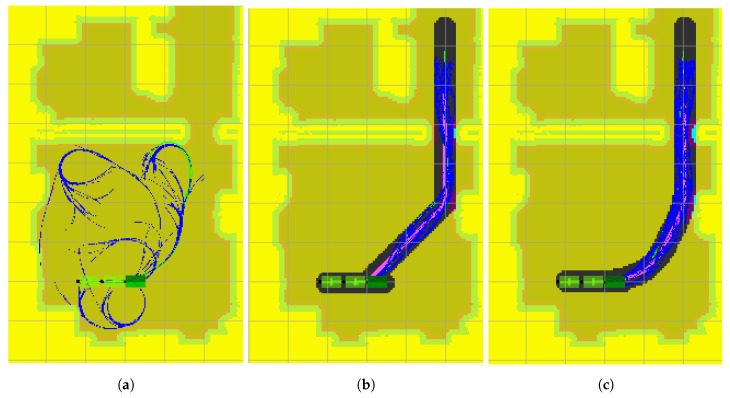
Goal 1 results. A small portion of total map (Figure 12) is presented. (**a**) Using RRT*; (**b**) Using ARA*-RRT*; (**c**) Using LA*-RRT*.

**Figure 14 sensors-20-06821-f014:**
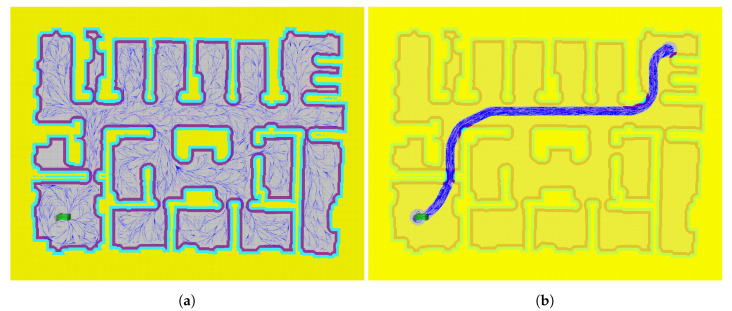
Path planning to travel between two corners of the map: (**a**) Path planning without mask (RRT*); (**b**) Path planning with mask (LA*-RRT*).

**Figure 15 sensors-20-06821-f015:**
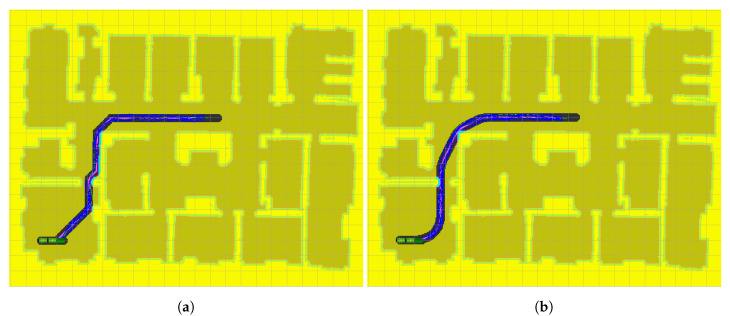
Goal 2 results: (**a**) Path planning with ARA*-RRT*; (**b**) Path planning with LA*-RRT*.

**Figure 16 sensors-20-06821-f016:**
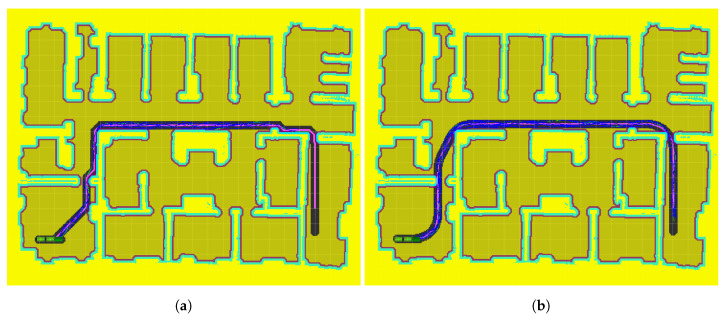
Goal 3 results: (**a**) Path planning with ARA*-RRT*; (**b**) Path planning with LARA*-RRT*.

**Figure 17 sensors-20-06821-f017:**
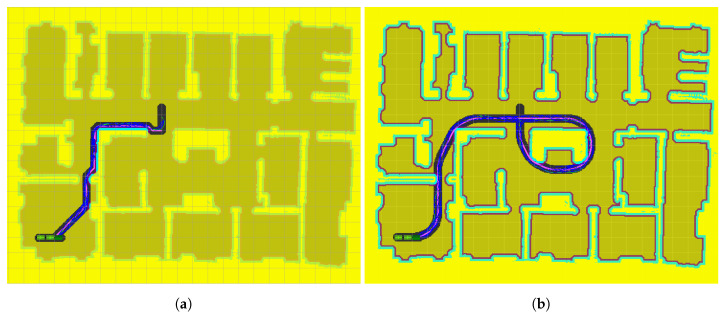
Goal 4 results: (**a**) Path planning with ARA*-RRT*; (**b**) Path planning with LA*-RRT*.

**Table 1 sensors-20-06821-t001:** Results.

		Time to First Solution (s)		First Solution Cost		Final Cost		Effectiveness
		**Mean**	**Std Dev**	**Mean**	**Std Dev**	**Mean**	**Std Dev**	
Goal 1	RRT*	N/A	N/A	N/A	N/A	N/A	N/A	0%
	ARA*-RRT*	4.99	3.24	14.88	0.95	12.99	1.18	100%
	LA*-RRT*	2.97	4.51	13.75	1.73	12.74	1.67	100%
Goal 2	RRT*	N/A	N/A	N/A	N/A	N/A	N/A	0%
	ARA*-RRT*	14.01	4.38	32.40	1.71	31.99	1.47	100%
	LA*-RRT*	14.48	7.91	33.90	3.34	33.09	2.71	100%
Goal 3	RRT*	N/A	N/A	N/A	N/A	N/A	N/A	N/A
	ARA*-RRT*	46.67	0.00	54.25	0.00	53.30	0.00	20%
	LA*-RRT*	30.37	4.44	60.54	0.68	60.20	1.00	100%
Goal 4	RRT*	N/A	N/A	N/A	N/A	N/A	N/A	N/A
	ARA*-RRT*	N/A	N/A	N/A	N/A	N/A	N/A	0%
	LA*-RRT*	29.30	6.57	57.62	6.33	57.12	6.67	100%

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
