# Peer review of "Efficient Path Planing for Articulated Vehicles in Cluttered Environments"

_sensors, 2020, doi:10.3390/s20236821_

Round 1

Reviewer 1 Report

The question in this paper is very clear and interesting, but the research method is not new, and there have been many related contents. The main idea of this work is to use RRT to find the optimal path in an "artificial corridor". For the map in this article, it is clear that the RRT is not available, so it is necessary to make the comparison with RRT. The conclusion section is not written in detail, the control and state quantities should be presented at the same time, and the vehicle and map parameters should be given. There are many problems in writing, including inconsistency between pictures and texts. For example, gamafk is in the text and gamakf is in the picture. Figure 7 does not indicate units...

Reviewer 2 Report

In general terms, the paper is well structured and well written.
It presents an algorithm for the motion planning of articulated logistic vehicles (e.g. tugger trains) that take into consideration also for the kinematics and dynamics of the vehicles. To improve the efficiency of the planners, in this work a kinodynamic discrete planning is considered.
The paper includes a theoretical formulation, a series of numerical tests using different mobile platforms and different trolleys to validate the method. A comparison is also made with other algorithms present in the literature.

Remarks:
- Fig 5 and 6. Add the reference frame in the figure.
- Fig 7. Add the measure units on axes.
- Row 320. Point at the end of the line is missing.
- Fig 9. Increase the font size and add the labels with the measure units.
- Reference [26]. Write in lowercase the conference venue.

Reviewer 3 Report

  1. Literature analysis

 Literature analysis very extensive. It covers both the benefits of automation and the evolution of computers and sensors, while the Authors are interested in algorithms. Therefore, the critical analysis of kinodynamic algorithms should be extended.

Notes to figures

Figures 1 and 2 should be placed in Chapter 2. Overview

Figure 3 adds nothing to the article, you can link it to Figure 1

Captions under figures (applies to figures 3, 4, 13, 14, 15, 16, 18).

All descriptions should be under the drawing, there should be references in the drawing, e.g. a), b)

e.g. "Figure 4. Trailers used in this research:  a - synchronized shafts trailer; b - single-shaft pallet trailer"

Figure 7. From the figure, it can be read that the course is linearized from the range from 60 to 140 in the description it is stated that from 60 to 120.

Figure 8 The figure does not show how many trailers there are and which trajectory the tractor will follow and which trailers will follow.

What's the difference between Fig. 16a and Fig. 17.? - This is a repetition

  1. Mathematical formula notes
  2. I believe that patterns should be numbered

For example

xË™k = vk ∗ cos (qk + gBk + p / 2)                                  (1)

line 230 in Figure 6 are not marked with gBk and gFk. Moreover, as it results from the dependence on gBk, it is a function of (a and b)

  1. Conclusions

Conclusions are a description of the results obtained. They lack a critical analysis aimed at showing the reasons for the imperfections of the analysed planners in terms of the topology of the distributed goals.

Round 2

Reviewer 1 Report

The manuscript still needs some improvements before it can be published.

  • A flow chart or structure diagram should be added to describe the algorithm clearly. First, a rough path is found based on A* algorithm; then an appropriate kernel size is determined through simulation based tests, the mask area is generated based the rough path and circular kernel; finally, the final path is obtained in the mask area through RRT*.
  • The proposed method is called LARA* -RRT*. However, A* algorithm is applied in fact, therefore, it is inappropriate to use the word ARA*(including the back of the LARA*, etc.). 
  • Since the kernel size is obtained by tests, the kernel size and train speed should be given.

Writing is not rigorous, for figure 6, phi-kf, phi-kb? or gama-kf, gama-kb? In the description before, gama and phi are two completely different angles.
